# Low-Cost Underwater Communication System: A Pilot Study

**Boguslaw Szlachetko** 

Faculty of Electronics, Photonics and Microsystems, Wroclaw University of Science and Technology, Wyb. Wyspianskiego 27, 50-370 Wroclaw, Poland; boguslaw.szlachetko@pwr.edu.pl

**Featured Application: The basic field of application of the presented communication modem is underwater robotics. The ultrasound modem would be potentially used to send simple commands and feedback from the robot as well as supporting the robot positioning task.**

**Abstract:** The aim of the paper is to present a simplified implementation of quadrature phase shift keying (QPSK) based underwater communication system. The presented solution addresses the problem of developing inexpensive, compact ultrasound modems able to be mounted on underwater robots. Simplifications introduced into the modulation and demodulation of QPSK signals do not disturb any parameter of the data link. The paper indicates that it is possible to realize modulation and demodulation on a simple microcontroller. Many hints are given on how to use hardware blocks embedded in a microcontroller, such as ADC, DMA, timers, etc. Experiments performed with the prototype modems allow to reach 4 kbps data rate on a distance of about 18 m.

**Keywords:** underwater communication; phase modulation; BPSK; QPSK; ultrasound transducer



## 1. Introduction

Underwater communication systems are constantly improved and developed. The vast majority of these systems are specially designed for large-budget projects. For this reason, the solutions used in these projects are not widely used, mainly due to the high price, patents, and other proprietary technology protection. It is beyond the scope of this article to discuss the meaningfulness of using these safeguards. However, the visible effect of the applied protection mechanisms is the limited access to knowledge and devices that allow for the widespread use of underwater communication. This document presents attempts to develop a device that can be constructed with a limited budget.

In the literature, one can find scientific articles that describe experimental underwater communication systems using commonly available elements [1,2]. The first paper describes an experimental multiagent system which utilizes autonomous underwater vehicles (AUV). The whole system is composed of AUVs, transponders, and sea buoys. All of these components have to communicate with each other. To achieve this aim, the ultrasound modems were specially designed in this project. Scientists developed an experimental communication technology called 2FSK. As it is stated by its name, double frequency shift keying modulation is utilized in each agent/modem. Two carrier frequencies, i.e., 35 and 38 kHz, are modulated separately, so a stream of two-bit symbols can be transmitted. The modem software works on PC104 computers with 64 MB of RAM, thus, the computing resources are quite large. Nonetheless, the system is designed to communicate at a rate of a few hundred bits per second. Therefore, a special communication protocol utilizing short 40-bit code words was designed.

The main goal of the other project [2] was to construct a compact ultrasound modem capable of communicating in an underwater environment. The modem was designed with the use of the ATmega128 microcontroller, so it can be powered by a 9 V battery. Waterproof plastic-shielded ultrasound transducers were chosen mainly because of their low price. The transducer resonant frequency is 30 kHz. The communication system is designed as

follows: bit '1' is transmitted using a carrier frequency of 30 kHz, whereas for bit '0', there is no signal at all, so it is a kind of amplitude modulation. Despite the simplicity of such a modulation scheme, the authors reported a quite high 1 kbps transmission rate, which is a good result. Unfortunately, this kind of modulation is not resistive to the multipath propagation of ultrasound waves. Similar work was reported in [3] but only short distance communication was reached.

A low power transceiver working with 200 kHz carrier frequency was deliberated in [4]. The authors reported the use of on–off keying (OOK) modulation with a 250 bps data rate, although the paper was focused on the design of an optimal amplifier, driving a piezoelectric transducer. Similar results were reported in [5]. The paper deliberated the possibility of implementing OOK modulation on the Cortex M3 microcontroller. The main advantage of this work was the use of a spherical piezoelectric transducer with a 70 kHz resonant frequency. The use of a spherical transducer allows for omni-directional transmission, which is an essential factor in underwater robotics. The paper reported a 1700 bps data rate for a short distance and 200 bps for a distance equal to 40 m.

In most underwater modem constructions, piezoelectric transducers are used. They are relatively easy to fabricate and tune their frequency parameters, so any resonant frequency can be chosen depending on the application. Additionally, a piezoelectric transducer can be simply modeled by its capacitance only [6]. Therefore, an amplifier circuit drives a voltage to charge and discharge this capacitance, so a class-D amplifier is a standard for this kind of construction [4]. Nevertheless, other transducers are taken into account in the research, such as a piezoelectric micro-machined ultrasound transducer (pMUT) [7,8]. This type of transducer is designed to work with the high coupling coefficient, so consequently, it can produce stronger output pressure, compared to traditional piezoelectric construction. Moreover, pMUT has an important property, namely, a small quality factor Q, which guarantees a wider bandwidth around the resonant frequency. Other transducers developed for underwater communication are structures based on piezocomposite materials. The main advantages of such technology are the light weight, low acoustic impedance, and broadband frequency characteristics. The last property is particularly important in the context of creating an efficient underwater communication network. Due to the very wide bandwidth of about 350 kHz with ripples of $\pm 5$ dB [9], one might think of using broadband modulation, traditionally reserved only for terrestrial telecommunications networks.

In the paper, a simplified approach is proposed. It means that the problems connected with an underwater propagation of ultrasound waves are ignored. Nevertheless, the literature on the subject indicates a few conditions for conducting experiments, where underwater propagation problems can be neglected. Very wide deliberations on ultrasound wave propagation were given in [10]. The authors pointed out that the noncoherent frequency modulation as well as phase modulation allows to avoid the difficulty of phase tracking in Doppler spread channels. In addition, they marked the possibility to utilize sophisticated modulations, such as orthogonal frequency division multiplexing (OFDM). OFDM signals, however, need to be transmitted by a broadband transducer because of their wide bandwidth. Other publications also investigate issues related to the propagation of ultrasonic waves. The theoretical limit of the underwater communication channel capacity was estimated to be 4 bits per second per Hz [11]. Consequently, for a symbol rate equal 1 kHz, it cannot be expected to achieve a baud rate greater than 4 kbps, regardless of the modulation used. Ultrasonic channel properties in the context of waveform propagation in the swimming pool can be found in [12]. Another environment of ultrasonic waveform propagation, namely, pipes, was analyzed in [13]. These research studies give important hints about the possible consequences of using underwater modems in such environments.

A separate class of underwater communication systems are solutions that try to mimic sea marine mammals, which use ultrasound in a natural way. Biologically inspired experiments are conducted in various directions. In [14], researchers focused on using a time-varying carrier frequency instead of the classical chirp spread spectrum technique to achieve the highest possible degree of similarity to a dolphin's whistle. The work [15]

explicitly indicates that the aim of this solution is hiding communication among naturally observed ultrasounds. A completely different approach is presented by the publication [16], where the authors focused on the development of the entire Morse alphabet composed of clicks, whistles, and songs emitted by humpback whales (Megaptera novaeangliae). All of these studies are extremely attractive; however, generating ultrasonic signals with a complex envelope and a time-varying carrier frequency requires large computational resources.

The last important factor directly related to this paper is the complication of the modem device. It means that one can select some sophisticated modulation techniques, but dedicated hardware, e.g., FPGA, is required [17–19]. A similar situation addresses the choice of an ultrasound transducer—a cheap off-the-shelf device is chosen for the implementation. On the other hand, a sophisticated broadband transducer, which is less common and usually protected by some propriety rights, would bring some improvement in power or bandwidth efficiency. Therefore, at the beginning of the project, one has to decide which factors are more important.

## 2. System Design

### 2.1. Initial Assumption

Using cheap off-the-shelf elements in the design is the base assumption in this project. According to this, the entire system should be built on the basis of a popular microcontroller, possibly a simple ultrasonic signal amplifier and a readily available piezoelectric transducer. It was also initially assumed that it would be necessary to use a floating point co-processor in the calculations. Therefore, it was decided to implement the modulation and demodulation algorithms in the software on the Cortex M4 microcontroller.

The assumed simplicity of the designed communication system limits the choice of modulation techniques because it was decided to use hardware timers embedded in the microcontroller to avoid the generation of cosine functions in the software. For this reason, binary phase shift keying (BPSK) and QPSK modulation were selected in the design. Other types of modulation, for example, those using chaotic functions [20], were also taken into consideration, but they require high computing power, and without hardware support, they are not possible to be implemented on popular microcontrollers.

Due to the pilot nature of the work, the communication protocol was not established. It was assumed that all tests would be performed with the use of two fixed nodes: the transmitter and the receiver. This simplification allows to avoid many problems related to access conflicts in the communication channel, typical for a multinode case.

### 2.2. Binary Phase Shift Keying (De)Modulator

The implementation of a cheap and simple modem for underwater communication began with BPSK modulation. It is the simplest form of phase modulation, in which the phase of the carrier signal can take two values shifted from each other by $\pi$.

$$x(t) = A \cos\left(2\pi f_c t + \phi(s)\right), \tag{1}$$

where $f_c$ is a carrier frequency, and phase $\phi(s) = \{0, \pi\}$ depends on the current symbol value $s = \{0, 1\}$. In practice it is equivalent to

$$
\begin{aligned}
x_0(t) &= A \cos\left(2\pi f_c t\right), \\
x_1(t) &= -A \cos\left(2\pi f_c t\right).
\end{aligned}
\tag{2}
$$

Carrier frequency $f_c$ for the system is determined by the resonant frequency of the ultrasound transducer. It allows for efficient transmission of the modulated waveform through the transducer. In the case of using a piezoelectric transducer, a cosine waveform can be easily replaced by the square waveform because together with some additional passive elements it can form a band-pass filter. The band-pass filter is necessary to suppress the higher order harmonic frequencies appearing in a square waveform spectrum.

The implementation of such modulation in software is extremely simple if the hardware timer is used, which is embedded in each microcontroller. The timer should be programmed to generate a square waveform on the output pin with frequency equal to $f_c$. Additionally, the counter should be configured to count the period of a symbol. After reaching the period, an interrupt is raised, so in the interrupt handling function, one can change the polarity of the output pin without touching any other register of the hardware timer. Consequently, all these steps can be executed in real time as necessary to implement the modulation procedure.

The demodulation procedure is much more complicated and time consuming. The analog-to-digital converter (ADC) embedded in the microcontroller has to be used. Thanks to the use of the direct memory access (DMA) subsystem, each sample taken by ADC is placed in a memory buffer. Each time the buffer is full, an interrupt is raised, and its handling function transfers the buffer content to a demodulation thread.

The demodulation of the BPSK signal needs to calculate the so-called in-phase ($I$) and quadrature ($Q$) components:

$$I(t) = r(t) * \cos{(2\pi f_c t)}, \tag{3}$$

$$Q(t) = r(t) * \sin{(2\pi f_c t)}, \tag{4}$$

where $r(t)$ is a signal received by the piezoelectric transducer. $I$ and $Q$ signals can be treated as traditional complex numbers, where the $I$ component is a real part and the $Q$ component is an imaginary part of a number. Therefore, the term IQ signal means a set of samples composed of $I$/real and $Q$/imaginary parts. This signal is amplified, filtered, and finally directed to the ADC. The forming of the IQ signal is executed in the discrete domain, i.e., an ADC's output discrete signal is multiplied by the cosine and sine functions and then low-pass filtered.

The demodulation thread calculates the IQ signal in the discrete domain. The values of the sine and cosine functions are tabulated prior to multiplication to speed up the calculation. Next, a low-pass filter is applied in both channels (i.e., $I$ and $Q$) and the integral over a period of a symbol is calculated. Finally, a decision about a symbol value is calculated. In the case of BPSK, only two possibilities exist, which are relatively easy to discriminate. One can see that this BPSK demodulation procedure works without synchronizing the sine/cosine function with the received carrier frequency. The choice of noncoherent demodulation is based on experiments. Phase locked loop (PLL) is considered. However, the real-time implementation in software is impossible. Simultaneously, the chosen Cortex M4 microcontroller does not have any PLL hardware block. Thus, an assumption is made that each transferred frame contains a priori known sync bits. These bits are used for detection of the start of the frame, as well as a training sequence. The training sequence is necessary to properly identify consecutive bit values in the decision block.

### 2.3. Quadrature Phase Shift Keying (De)Modulator

Due to the need to increase the bandwidth efficiency, this paper touches the implementation of the QPSK communication system on the Cortex M4 microcontroller. The bandwidth efficiency of $M$-state PSK signals can be calculated according to the formula

$$\rho = \frac{\log_2{M}}{2}, \tag{5}$$

where $M = 2$ for the BPSK signal and $M = 4$ in the case of the QPSK signal. The bandwidth efficiency is defined as the ratio between the transmission bit rate and frequency bandwidth of an MPSK signal. Therefore, the higher $M$ value is taken, the higher bandwidth efficiency is reached, but at the cost of higher complication of the decision block. Therefore, the choice of QPSK modulation is a kind of trade-off between the bandwidth efficiency of a transmitted signal and complications in the software, which implements modulation and demodulation, using the limited resources of the microcontroller.

In theory, the QPSK modulation signal is defined according to (1) with four possible phase values $\phi(s) = \{0, \pi/2, \pi, 3\pi/2\}$.

The block diagram of the modulation procedure is shown in Figure 1. The information intended for transmission is converted into a bit stream, which is then divided into two streams, namely, odd and even bits. Next, these bits are used to control the hardware timer of the Cortex M4 processor. Similar to the BPSK modulation process, the hardware timer runs at the carrier frequency $f_c$, which is also the resonant frequency of the ultrasound transducer used for transmission.

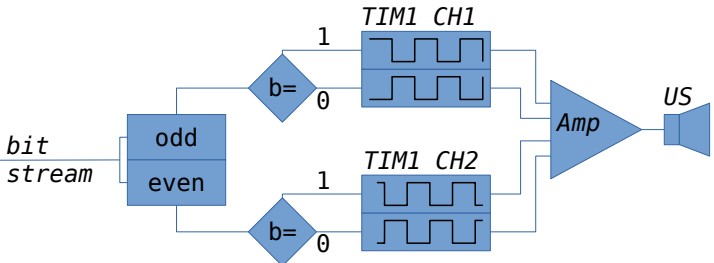

**Figure 1.** The diagram of QPSK modulation process implemented with the use of hardware timers.

It is worth noting that the Cortex M4 processor does not have enough computing power necessary for the software generation of sine wave and switching its phase in real time. It is because the microcontroller has to process lots of other information, and not only calculate a consecutive value of sine wave at consecutive time instants. The built-in floating point co-processor also does not solve this problem. Thus, the traditional implementation of QPSK modulation using only software is not possible at all.

Software implementation of the QPSK modulation procedure is only slightly more complicated compared to the BPSK modulation procedure. The hardware timer is programmed to generate a square waveform at the output pin, and a counter counts the period of the symbol. At the end of the symbol's period, an interrupt is raised. Its handling routine decodes two-bit symbols, i.e., extracts the odd and even bits and configures the hardware timer by changing the output compare register value. Thus, together with checking some condition values and clearing the respective interrupt flags, the whole interrupt routine is quite short and does not disturb the real-time processing of the modulation procedure.

Regarding Figure 1, the output pin of the hardware timer is connected to an amplifier and then directly to the piezoelectric transducer. It was mentioned that the final stage of the amplifier contains some passive elements which, together with the piezoelectric transducer, play the role of a band-pass filter. This part of the project, however, is beyond the scope of the paper.

The demodulation process of a received QPSK signal is also more complicated when compared to the BPSK. In Figure 2, the simplified idea of the QPSK demodulation is shown. First, the received QPSK signal is sampled and multiplied by the cosine and sine function to obtain the IQ signal. Next, a low-pass filter is applied and the integral is calculated over the period of a symbol. The final block is a decision block. Comparing this procedure with the BPSK demodulation, it has to discriminate between the four possible states. Moreover, because PLL is not implemented, the decision block has to properly guess which state represents a particular symbol.

In Figure 3, one can see a constellation diagram. It shows the integrated values of the IQ signal, where $k \in \{0, \ldots, 127\}$ means the consecutive number of a symbol. The data come from hardware-in-the-loop (HITL) simulations. First, the QPSK modulator part together with the channel and receiving transducer is simulated on PC computer. Then the resulting signal is injected into the microcontroller memory just after the ADC block (look at the Figure 2. The rest of the demodulation process is run on the microcontroller. After completing a frame of data—in this case, 128 symbols—the data are transferred through the debug channel and stored on the local disk of PC computer. Summarizing,

in Figure 3, 128 points with value equal to $syb(k) = I(k) + iQ(k)$ are shown. Points are grouped into four subsets, i.e., four states, which are relatively easy to distinguish. Each subset represents one of four possible two-bit symbols, namely, 00, 01, 11, and 10. Looking at this constellation, no one knows which group of points represents a particular symbol. It is because the internal carrier frequency generated is not synchronized with the carrier frequency of the received signal. This kind of demodulation is called a noncoherent demodulation technique. To overcome the synchronization problem, a training sequence is utilized. The training sequence is composed of four unique symbols transmitted with a constant order at the beginning of each frame. Thanks to the known order, the decision block can "learn" the placement of each symbol on an IQ plane, and next, it can properly decode the information from the received frame.

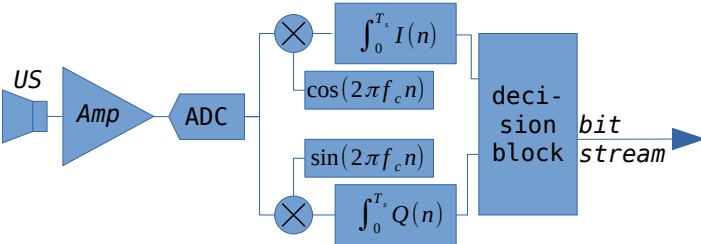

**Figure 2.** The diagram of QPSK demodulation process. A traditional demodulation algorithm with the use of IQ signals is utilized.

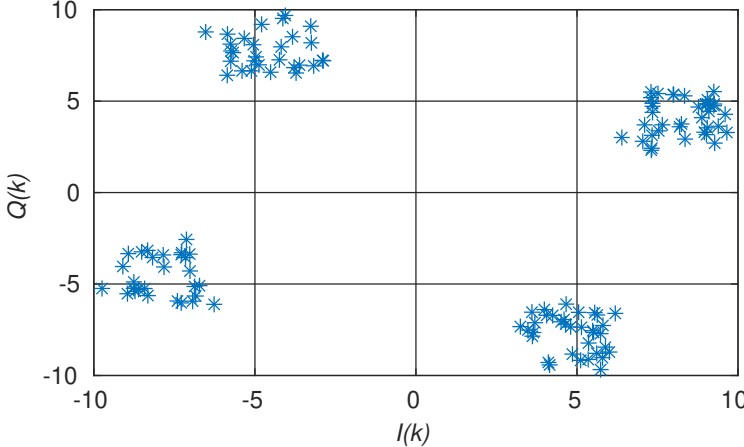

**Figure 3.** A constellation of integrated IQ signal of a received frame consisting of 128 symbols modulated with the speed of 2 kbps.

To completely describe the QPSK demodulation thread, it is worth noting that the demodulation thread constantly waits for the next buffer of information given by the DMA interrupt routine. Remember that DMA gathers consecutive samples from ADC into the buffer. After receiving a semaphore from the DMA interrupt routine, the demodulation thread copies the buffer into the internal memory and next checks in the buffer to see if there exists a carrier frequency or not. In the case of a positive check, the thread raises an internal flag informing that the start of the frame has been detected. As a consequence, the demodulation procedure is started, which means that the internal carrier frequency generator starts to generate a cosine/sine waveform. Indeed, this way, carrier frequency synchronization cannot be guaranteed. However, it was mentioned that it is not necessary in this project.

### 3. Conditions for Conducting the Experiment

A working prototype of the modem was tested in a river port basin. It is a dead end of the Odra river; hence, it can be assumed that the water at this point is stagnant. A situation plan of the performed experiment is shown in Figure 4.

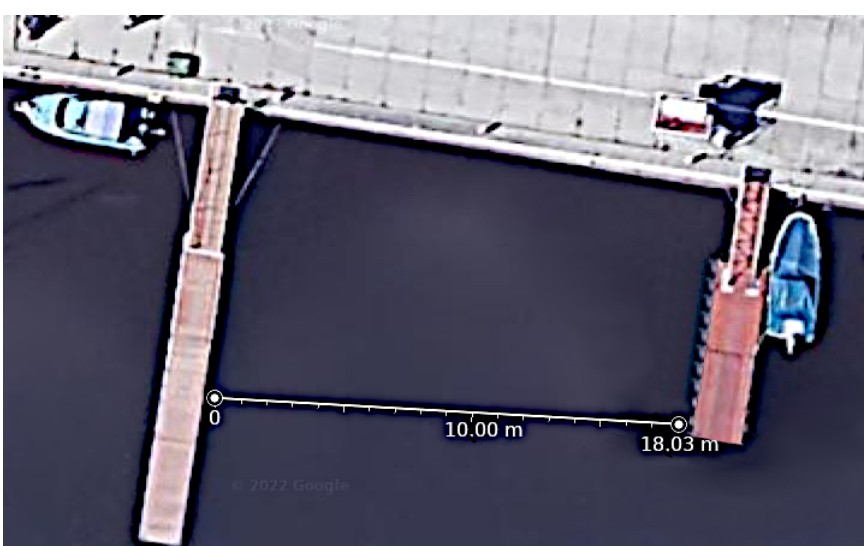

**Figure 4.** Map of the place of the experiment, taken from Google Maps. Coordinates: 51°08′06.0″ N 16°59′57.9″ E.

During the day of the experiment, there were no boats around platforms. Therefore, the direct path of an ultrasound wave was not disrupted. The depth in this place varies between 2 and 4 m. The distance between the platforms is about 18.2 m according to Google Maps. The exact knowledge about the bottom is not available, but an assumption was made that it is unevenly rocky and covered with years of mud and vegetation.

For this project, an off-the-shelf ultrasonic transducer was chosen. It is a quite popular model, TD0040, branded by Hurricane, a Chinese company. In Figure 5 the frequency characteristic of the sound pressure level (SPL) is shown. One can easily find it on the internet. Additional important parameters of this particular transducer are the beam width 50°, high power, and IP67 housing. In the project, a small voltage which drives the transducer—only 15 [V]—does not utilize the available power of the transducer. Nevertheless, it is a pilot study, so the delivered power is high enough to test the modem.

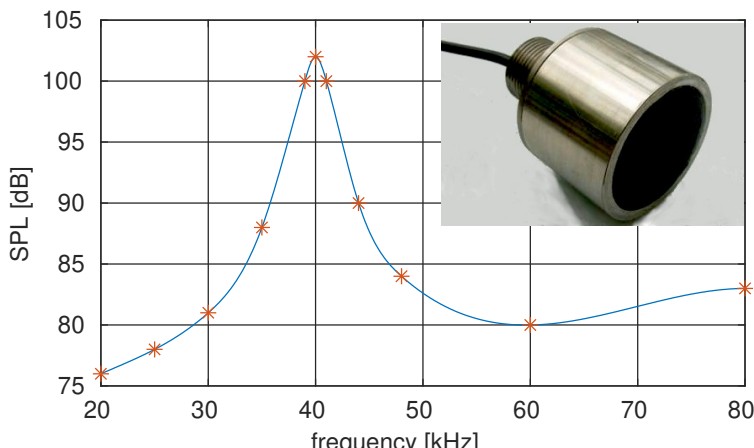

**Figure 5.** A photo and frequency characteristic of the sound pressure level of the ultrasonic transducer used in the project.

## 4. Results

During the first phase of the experiment, the prototype of the modem was tested with the use of BPSK signals and a number of different symbol speeds. Additionally, the signal received by the transducer was recorded on a digital oscilloscope.

The result is presented in Figure 6. The start of the frame can be seen. In addition, the moment of phase change in the BPSK signal is evident. The speed of the transmission was set up to 1000 symbols per second, which gives 1000 bps data link. In Figure 6, a high level noise is visible. Despite this, the modems were able to communicate.

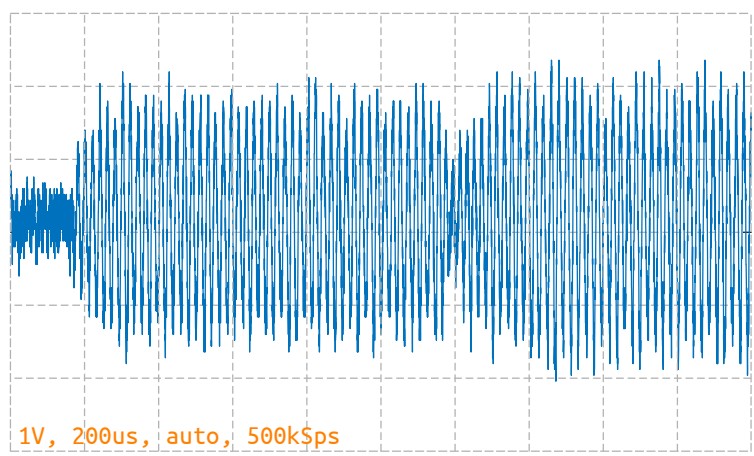

1V, 200us, auto, 500k$ps

**Figure 6.** Received BPSK signal registered at the output of the amplifier. Signal registered with the Tectronics oscilloscope sampling at 500 kHz.

In the next Figure 7, the IQ signal of the QPSK modulation is shown. In this particular trial, the modulation speed is 2000 symbols per second, which gives a 4000 bps transmission rate. The observed noise slightly influences the IQ signal, but the integral calculated over the symbol period gives a mean value. Thus, the noise is suppressed and finally a right decision about the value of a symbol is possible.

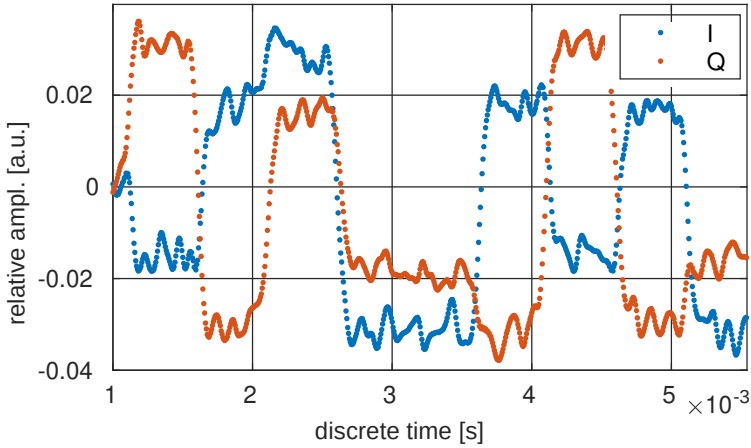

**Figure 7.** IQ signals before integration taken from the microcontroller thanks to the HITL simulations.

In theory, the frequency band of the QPSK signal can be found by

$$B = \frac{2R_b}{\log_2 M},$$ (6)

where $R_b$ is a symbol rate, and $M$ is a number of possible phase values utilized by a modulation. In the case of QPSK modulation, $M = 4$. The bandwidth can be taken from

the ultrasound transducer characteristics—see Figure 5 $B \approx 2400$ for the standard 3 [dB] level. Therefore, one can find the theoretical maximum symbol rate according to

$$R_b = \frac{B \log_2 M}{2} \approx 2400 \text{ [symb/s]}. \tag{7}$$

In the experiment, setting up a symbol rate higher than 2000 [symb/s] resulted in some transmission errors. It is probably so, because too small power was emitted by the ultrasound transducer. The power of the ultrasound signal can be increased by increasing the voltage, but this requires a modification of the charger circuit.

## 5. Discussion

The paper presents the results of a pilot study in which the theoretical and practical aspects of phase—particularly QPSK—modulation and demodulation were developed. Modem devices consisting of an ultrasound transducer, amplifier and power supply circuit and the popular Cortex M4 microcontroller were designed and tested. The feasibility studies have shown the possibility of creating an inexpensive modem for applications of underwater communication. Such modems can be very attractive in the field of underwater robotics. In the proof-of-concept experiments, 4000 bps speed was reached. This link can be utilized by the robots to send simple commands, status, position, etc. Solutions known from the literature describe lower transmission speeds compared to the rate reached in this project [1,4,5]. Moreover, each transmitted frame can be treated as an ultrasound probing pulse, so it can deliver information about obstacles around the robot. This is a classic echolocation approach, but at this stage of the project, it is only a potential possibility of the system. The second interesting property of this design is the low level of complexity of the device and the fact that the processor still has free computing power that can be used. For example, it can be used to control the actuators or for the calculation of the attitude and positions of the robot. Other designs require the use of complex technologies, such as FPGA [17–19], which disqualify them in popular robotic projects.

The conditions of the experiment were relatively good. It means the modems together with ultrasound transducers were submerged at a depth of about 1.5 m. There were no obstacles around (just two yachts, moored far away from the platforms), and the water was almost stagnant. The author is aware of many problems related to the propagation of ultrasonic waves in shallow water. Due to the preliminary nature of the project, this issue was not addressed in this paper. However, future works are planned to test the immunity of QPSK modulation to multipath propagation. Another unsolved issue is the influence of the speed of flowing water (alternatively of a moving object with an ultrasonic modem installed on it) on the capability of demodulation of the received transmission. It is well known that the Doppler effect would change the wave carrier frequency [10], and therefore the question remains of whether a simplified implementation of a PLL-free demodulator will be sufficient.

**Funding:** This research received no external funding.

**Institutional Review Board Statement:** Not applicable.

**Informed Consent Statement:** Not applicable.

**Conflicts of Interest:** The author declares no conflict of interest.

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
