# Peer review of "Low-Cost Underwater Communication System: A Pilot Study"

_applsci, doi:10.3390/app12073287_

Round 1
Reviewer 1 Report
The paper reports a software-hardware solution for an easy-to-assemble underwater communications system made from off-shelf components. The practical importance of the reported study follows from the low availability of commercial-grade underwater communication systems, which slows down the development of wireless UAVs and underwater robots. Despite the results shown in the paper being relatively modest, I believe the paper can significantly contribute to developing hydroacoustic engineering solutions. My key concern is the scientific novelty of the paper. Please, emphasize it in the revised manuscript.
Please, find my other questions below.
1. Is the proposed underwater communication scheme robust to the crosstalk issue? How the reflections from bottoms\shore\walls of the test basin are handled?
2. What if several such systems work simultaneously? How can channel separation be organized? Are there some built-in algorithms to change the carrier frequency in case of conflict?
3. State-of-the-art solutions in underwater communications systems suggest using biological signals (e.g., whales singing or dolphin's pings) because these signals are best-suited for traveling underwater with minimal distortions. Have you compared your approach with the abovementioned technique?
4. What about chaotic signals? Can chaos-based modulation provide a higher speed of transmission/bitrate? Using chaotic generators, for example, based on simple discrete chaotic maps, can also reduce the crosstalk effects. Recently, many novel types of modulation have been provided for chaotic signals, e.g., symmetry-based modulation technique.
5. Some photos of the experimental testbench or figure illustrating the scheme of the experiment (distance, depth, etc.) could clarify the conditions of the experiments. I believe it is essential for such a field study.
6. "The issue of the propagation of ultrasonic waves under water is beyond the scope of this article" - I recommend removing this sentence. Every engineer working in hydroacoustics inevitably ends with studying this issue. Maybe, it is worth changing to "we use the simplified approach in the current study"
7. "Moreover, each transmitted frame can be treated as an ultrasound probing pulse, so can deliver information about obstacles around the robot" - I recommend clarifying this phrase. Does it mean passive echolocation using communications system pulses?
8. "costly hardware, like, e.g., FPGA chips" - First, I recommend removing "like" or "e.g." to avoid tautology. Second, simple FPGA chips are not "costly hardware" at all and are broadly used in hydroacoustics.
9. What was the sound environment in the "river port basin"? What was the bottom type - soft, solid, gravel, stones, mud? Please, report more about the experiments' conditions.
10. I recommend inverting colors in Fig.5 to avoid a black background.
11. The manuscript is to be extensively proofread. Many typos, including stylistic and grammar mistakes, can be observed.
Author Response
Comments and Suggestions for Authors
The paper reports a software-hardware solution for an easy-to-assemble underwater communications system made from off-shelf components. The practical importance of the reported study follows from the low availability of commercial-grade underwater communication systems, which slows down the development of wireless UAVs and underwater robots. Despite the results shown in the paper being relatively modest, I believe the paper can significantly contribute to developing hydroacoustic engineering solutions. My key concern is the scientific novelty of the paper. Please, emphasize it in the revised manuscript.
Please, find my other questions below.
Thank you very much for your suggestions. I answered all of them, and the paper has been improved thanks to your revision.
1. Is the proposed underwater communication scheme robust to the crosstalk issue? How the reflections from bottoms\shore\walls of the test basin are handled?
No. The system was not tested on the crosstalk issue. I mean that despite the existence of a crosstalk effect in the experiment, the modem has been able to receive and demodulate the signal. Of course, further research should be focused on this issue. Some explanation is added in the paper - line 74.
2. What if several such systems work simultaneously? How can channel separation be organized? Are there some built-in algorithms to change the carrier frequency in case of conflict?
No. At this stage of development, the problem of simultaneous communication was not addressed. The transducer utilized in the project does not allow to switch to the other carrier frequency. Therefore, only the time division scheme can be applied to resolve the conflicting situation. In future work, the use of the wideband transducer is considered. Some explanation is added in the paper - line 127
3. State-of-the-art solutions in underwater communications systems suggest using biological signals (e.g., whales singing, or dolphin's pings) because these signals are best-suited for traveling underwater with minimal distortions. Have you compared your approach with the above-mentioned technique?
No. I do not consider using pulses inspired biologically. These kinds of signals are wideband, while a relatively narrowband transducer was utilized in the project. But I've added a paragraph in the Introduction - lines 92-103
4. What about chaotic signals? Can chaos-based modulation provide a higher speed of transmission/bitrate? Using chaotic generators, for example, based on simple discrete chaotic maps, can also reduce the crosstalk effects. Recently, many novel types of modulation have been provided for chaotic signals, e.g., symmetry-based modulation technique.
A chaos-based modulation technique has not been considered. Mainly due to the use of hardware timers embedded in the micorocontroller. However, it can be a good direction for further research. Please look at line 124.
5. Some photos of the experimental testbench or figure illustrating the scheme of the experiment (distance, depth, etc.) could clarify the conditions of the experiments. I believe it is essential for such a field study.
Yes. I agree. Figure 4 provides such information. More explanation can be found in the section "3. Conditions for conducting the experiment"
6. "The issue of the propagation of ultrasonic waves underwater is beyond the scope of this article" - I recommend removing this sentence. Every engineer working in hydroacoustics inevitably ends with studying this issue. Maybe, it is worth changing to "we use the simplified approach in the current study"
Yes. I agree. The sentence is changed.
7. "Moreover, each transmitted frame can be treated as an ultrasound probing pulse, so can deliver information about obstacles around the robot" - I recommend clarifying this phrase. Does it mean passive echolocation using communications system pulses?
No. I have in mind a classical active echolocation algorithm. It means that a modem emitting a frame of information adds a unique header at the beginning of the frame, so the recipient can identify the id of the transmitter, timestamp/sequence number, etc. However, the transmitter can receive the echo of its own frame, and thanks to the information in the header, it can calculate the distance. Such a system is, of course, very simple. At this stage of the project, the transmission protocol is not fixed yet, thus, the possibility of echolocation is only potential. Please look at line 293.
8. "costly hardware, like, e.g., FPGA chips" - First, I recommend removing "like" or "e.g." to avoid tautology. Second, simple FPGA chips are not "costly hardware" at all and are broadly used in hydroacoustics.
Thank. I understand. I wrote "costly" but I have in mind an additional cost of hardware and software development, not only the price of FPGA chips. Probably, further work will be supported by a national grant, therefore, FPGA technology will be taken into deliberation.
9. What was the sound environment in the "river port basin"? What was the bottom type - soft, solid, gravel, stones, mud? Please, report more about the experiments' conditions.
I have no exact knowledge about the bottom. I can only assume that the bottom was unevenly rocky and covered with years of mud and vegetation. This assumption is based on the fact that it was an older part of the harbor that had not been used for a long time. I agree with your suggestion to add more information about the experiments' conditions. Please look at the section "3. Conditions for conducting the experiment".
10. I recommend inverting colors in Fig.5 to avoid a black background.
Yes. It is a good idea. I've changed the Figure.
11. The manuscript is to be extensively proofread. Many typos, including stylistic and grammar mistakes, can be observed.
OK. I will do my best.
Reviewer 2 Report
It seems to me that the author attempts to develop a low cost underwater communication system. However, the approach is not convincing.
(1) Why FPGA is costly / the use of FPGA is complex? (2) Are broadband transducers really hard to buy? (3) Why did the author not select a more powerful processor if Cortex M4 does not have enough computing power? (4) What is reason that the author does not adopt the PLL solution, but decides to utilize a training sequence? (5) Where doe come from the I/Q signal constellation, simulation or measurement?
If the communication system cost really poses a problem to an underwater device in the author's country, which I strongly doubt, the paper shall not be submitted to an international journal.
Besides, the writting is defficient, for instance the resonant frequency "are" 30 KHz, for a distance "equal" 40 m, done in a "real-time", in "the" Figure 1, throughout the paper "in this project", it "worth to".
Author Response
# Comments and Suggestions for Authors
It seems to me that the author attempts to develop a low-cost underwater communication system. However, the approach is not convincing.
If the communication system cost really poses a problem to an underwater device in the author's country, which I strongly doubt, the paper shall not be submitted to an international journal.
Besides, the writting is defficient, for instance the resonant frequency "are" 30 KHz, for a distance "equal" 40 m, done in a "real-time", in "the" Figure 1, throughout the paper "in this project", it "worth to".
Thank you very much for your valuable suggestions. I have addressed your questions in the paper. According to the language skills - all mistakes and inconsistency has been removed from the text.
1. Why FPGA is costly / the use of FPGA is complex?
Thank you for this question. I wrote "costly" but I have in mind an additional cost of hardware and software development, not only the price of FPGA chips. Probably, further work will be supported by a national grant, therefore, FPGA technology will be taken into deliberation.
2. Are broadband transducers really hard to buy?
I agree. I wrote "hard to buy" but I meant that wideband transducers are much less common than narrowband ones. In the design, the assumption was that the modem should be made up of widely available modules, therefore, it was decided to use a popular narrowband transducer. Please look at lines 114-119.
3. Why did the author not select a more powerful processor if Cortex M4 does not have enough computing power?
The reason is the same as according to the choice of a popular narrowband ultrasound transducer. The Cortex M4 was more popular and widely available than Cortex M7, thus The Cortex M4 was chosen. Regardless of this choice, a more powerful Cortex family processor can be used at any time. Changing the microcontroller to a more advanced microprocessor will entail changing many assumptions related to power supply, cooling system, modem space limitation, etc. I have added subsection "2.1 Initial assumptions" to explain the reason why the Cortex M4 was chosen.
The reason is the same as according to the choice of popular narrowband ultrasound transducer. The Cortex M4 were more popular and widely available than Cortex M7, thus The Cortex M4 for were chosen. Regardless of this choice, a more powerful Cortex family processor can be used at any time. Changing the microcontroller to a more advanced microprocessor will entail changing many assumptions related to power supply, cooling system, modem space limitation, etc. I have added subsection "2.1 Initial assumption" to explain the reason why the Cortex M4 was chosen.
4. What is reason that the author does not adopt the PLL solution, but decides to utilize a training sequence?
According to my knowledge microcontrollers from Cortex M3/M4/M7 family do not contain hardware components supporting the PLL implementation. I mean that there exists the PLL embedded in the microcontroller but it is only used for trimming the system clock. Therefore, approximation has been made that the computational cost of software implementation of PLL is too high because about 50% of computational time should be left for robotic parts of the software. Some discussions are provided in lines 295-298
5. Where doe come from the I/Q signal constellation, simulation or measurement?
IQ constellation presented in Figure 3 comes from simulation. Without PLL the constellation is rotated by the unknown phase, which can be observed in Figure 3. I've added an explanation in the paragraph starting in line 218.
Reviewer 3 Report
A Multicarrier system with two subbands is presented. The document has no structure. Example: The experiment.
line 225, 239: In the abstract and the final section "discussion" you can find something about the experiment. "Experiments done with the prototype modems allows to reach 4 kbs data rate on a distance about 18 m."
In the discussion the experiment "The conditions of the experiment were relatively good. It means the modems together with ultrasound transducers were submerged at depth about 1.5 m, ... The author is aware of many problems related to the propagation of ultrasonic waves in shallow water." is descibed, but a experiment section inside the paper is missing. The reader needs the complete experiment description when, where, conditions, environent in a own section. What is the scenario, the topology at the river? Soundspeed, Current, water depths?
Difficult to read: Words like "a" and "the" are missing,
With the title "A Software Modem for Underwater Communication System: A Pilot Study" the reader expect a software defined modem presentation. But standard parts of a software modem are missing - the structure
of the document should follow this.
Line 77 search for "Orthogonal Frequency Division Multiplexing (OFMD)" set "Orthogonal Frequency Division
Multiplexing (OFDM)"
line 147 search "bits per second per Hertz unit." set "bit unit." (per second per Hertz = 1)
The reader needs a reference to compare the results. Please use the Watermark benchmark
(https://www.ffi.no/en/research/watermark) to test your system, Parameters like signal to noise ratio (SNR), should plot over time. Display the simulation results.
Author Response
Comments and Suggestions for Authors
Thank you very much for your valuable suggestions. I have read and corrected each one carefully. I really appreciate the Watermark system you pointed out.
1. line 225, 239: In the abstract and the final section "discussion" you can find something about the experiment. "Experiments done with the prototype modems allows to reach 4 kbs data rate on a distance about 18 m." In the discussion the experiment "The conditions of the experiment were relatively good. It means the modems together with ultrasound transducers were submerged at depth about 1.5 m, ... The author is aware of many problems related to the propagation of ultrasonic waves in shallow water." is described, but a experiment section inside the paper is missing. The reader needs the complete experiment description when, where, conditions, environment in a own section. What is the scenario, the topology at the river? Sound speed, Current, water depths?
I agree. The section " Conditions for conducting the experiment" is added to the paper together with a figure and explanation of the experiment's conditions.
2. Difficult to read: Words like "a" and "the" are missing,
I agree. I hope - all mistakes have been corrected.
3. With the title "A Software Modem for Underwater Communication System: A Pilot Study" the reader expect a software defined modem presentation. But standard parts of a software modem are missing - the structure of the document should follow this.
I understand your point. Partially, Figure 2 presents a structure. But you have right - in the paper, only the modulator/demodulator part is described. Therefore, I decided to slightly change the title of the paper.
4. Line 77 search for "Orthogonal Frequency Division Multiplexing (OFMD)" set "Orthogonal Frequency Division Multiplexing (OFDM)"
I agree. Corrected.
5. line 147 search "bits per second per Hertz unit." set "bit unit." (per second per Hertz = 1)
Yes. I have right - bit/s/Hz=bit. But I wanted to point out the physical meaning of this unit. Finally, I decided to remove this part of the sentence.
6. The reader needs a reference to compare the results. Please use the Watermark benchmark
(https://www.ffi.no/en/research/watermark) to test your system, Parameters like signal to noise ratio (SNR), should plot over time. Display the simulation results.
Unfortunately, the Watermark offers no channel with a bandwidth around 40kHz. But I want to assure you that I really appreciate your tip to use the Watermark simulator. Obtaining funds from the national grant is planned, so I hope that we can establish closer cooperation in the field of underwater communication systems.
Round 2
Reviewer 1 Report
Thank you for providing a revised version of your interesting paper. Despite I still believe the reported study is purely engineering, I like it due to its great practical value. Indeed, many researchers and engineers lack cheap underwater communications systems for their submersible drones. The reported study provides such a solution in detail, which makes it definitely worth publishing.
I recommend proofreading the manuscript a bit further before. Maybe some spellcheck services will be handy here. I wish the Author any success in his further research.
Author Response
Proofreading has been done. I really appreciate your valuable comments. Thank you very much for your support.
Reviewer 2 Report
The font size in Figure 1, 2 shoud be reduced.
Author Response
OK. I have changed the font size inside all Figures.
Thank you for your valuable comments and your support.
Reviewer 3 Report
Our points are not answerd. Experiment description, sound speed, echo structure, channel impulse response? Figures results of simulations or measurements? Please rewrite your paper.
Author Response
The experiment condition is described in a separate section, i.e. Section 3. There is no information about the mentioned aspects like sound speed, echo structure, channel impulse response, and so on because it is a pilot study. To investigate the influence of these aspects on the quality of the proposed communication system one has first to be sure that all necessary blocks - hardware and software - work properly.
Regarding the question about figures - I have added some explanations directly to figure captions. Please look at Figure 3 and Figure 7 captions. Also, there is some additional information provided in the paragraph starting at line 218.
Further work is planned. Another transducer, which uses a carrier frequency close to 37 kHz will be chosen and software will be slightly changed to be able to work in a band of 37 kHz. This way it would be possible to simulate the proposed system using the Watermark benchmark.
Thank you very much for your valuable suggestions and your help.